# Large-scale molecular endotype discovery in synovial fluid reveals osteoarthritis as a single biological continuum

T. A. Perry [1,21] ✉, Y. Deng[1,21], P. A. Hulley [2], R. A. Maciewicz [1], J. Mitchelmore[3], S. Larsson [4], J. Gogain[5], S. Brachat[3], A. Struglics [4], C. T. Appleton [6], S. Kluzek [2,7], N. K. Arden [2,8], D. Felson[9], L. Bondi[10], M. Kapoor [11], L. S. Lohmander [4], T. J. Welting [12], D. A. Walsh[13,14], A. M. Valdes [13], Luke Jostins-Dean [1,22], Fiona E. Watt [1,15,22], B. D. M. Tom [10,22] & T. L. Vincent [1,22] ✉ On behalf of the STEpUP OA Consortium*

Knee osteoarthritis affects 40% of people during their lifetime, significantly impacting societies worldwide. Its molecular pathogenesis remains poorly understood and variable clinical phenotypes suggest it may be more than one disease. We established Synovial fluid To detect Endotypes by Unbiased Proteomics in OA (STEpUP OA) to search for molecular endotypes in knee OA synovial fluid, and to reveal key pathobiological pathways across 1361 individuals with knee OA. Using unsupervised clustering, a single cluster representing a biological continuum is observed, primarily driven by "Epithelial Mesenchymal Transition". Distinct molecular endotypes are not detected. "Angiogenesis", "Complement" and "Coagulation" are enriched for after stratification by clinical phenotype (obesity status, biological sex). Complement and coagulation are associated with the inflammatory marker, C-reactive protein. Associations with patient-reported knee pain are weaker. These findings support knee OA as a biological continuum, identify common and phenotype-enriched targetable pathways, and a rationale for stratification in clinical trial design.

Osteoarthritis (OA) of the knee is common, affecting up to a third of adults aged 60 years or older[1]. Characterised by failure of the synovial joint, OA is a major contributor to healthcare costs and is a leading cause of disability, largely through chronic pain and limitations in function. Age and obesity are important risk factors, both of which have contributed to increasing disease burden across global populations[2-4]. There are currently no approved treatments for knee OA that effectively target structural disease and those that target

---

[1]Centre for Osteoarthritis Pathogenesis Versus Arthritis, Kennedy Institute of Rheumatology, NDORMS, University of Oxford, Oxford, UK. [2]Nuffield Department of Orthopaedics, Rheumatology, and Musculoskeletal Sciences, University of Oxford, Oxford, UK. [3]Novartis Biomedical Research, Basel, Switzerland. [4]Faculty of Medicine, Department of Clinical Sciences Lund, Orthopaedics, Lund University, Lund, Sweden. [5]Standard BioTools (previously known as SomaLogic), Boulder, Colorado, USA. [6]Department of Medicine, University of Western Ontario, London, Ontario, Canada. [7]NIHR Nottingham Biomedical Research Centre and Versus Arthritis Sport, Exercise and Osteoarthritis Centre, University of Nottingham, Nottingham, UK. [8]Centre for Sport, Exercise and Osteoarthritis Research Versus Arthritis, University of Oxford, Oxford, UK. [9]Section of Rheumatology, Boston University School of Medicine, Boston, Massachusetts, USA. [10]MRC Biostatistics Unit, University of Cambridge, Cambridge, UK. [11]Schroeder Arthritis Institute, University Health Network, Toronto, Ontario, Canada. [12]Laboratory for Experimental Orthopedics, Department of Orthopedic Surgery, Maastricht University, Maastricht, Netherlands. [13]Pain Centre Versus Arthritis, Advanced Pain Discovery Platform, and the NIHR Nottingham Biomedical Research Centre, University of Nottingham, Nottingham, UK. [14]Sherwood Forest Hospitals NHS Foundation Trust, Sutton in Ashfield, UK. [15]Department of Immunology and Inflammation, Imperial College London, London, UK. [21]These authors contributed equally: T. A. Perry, Y. Deng. [22]These authors jointly supervised this work: L. Jostins-Dean, F. E. Watt, B. D. M. Tom, T. L. Vincent. *A list of authors and their affiliations appears at the end of the paper. ✉e-mail: thomas.perry@kennedy.ox.ac.uk; tonia.vincent@kennedy.ox.ac.uk

symptomatic disease have modest efficacy and are associated with adverse events[5,6]. There remains, therefore, a major unmet clinical need.

Limited understanding of disease pathogenesis coupled with a failure to translate findings from basic research to clinical settings has hampered clinical translation in OA[7,8]. Another significant challenge is the broad clinical spectrum of disease that has led many to question whether OA is one disease, or whether it is driven by multiple different pathways that converge on a common joint pathology[9,10]. Multiple clinical phenotypes have been suggested in the literature[11–13], but these have not been validated as clinically useful stratification tools either when testing treatment responses or as predictors of disease progression[14–16]. Endotypes, defined by distinct molecular signatures, may have higher value, and could in part explain observable characteristics of a phenotype[17]. This is an important hypothesis that has never been formally assessed.

Recent advances in understanding complex disease have been greatly enhanced by the application of multi-omic approaches to disease-relevant tissues[11,18]. The strengths of these approaches are the focus on human disease cohorts at scale, the unbiased and systematic nature of molecular identification, the ability to map molecules to shared pathways, and the ability to replicate results across independent cohorts. Technological advances in genomics, transcriptomics, and proteomics have enabled such studies to be carried out with low tissue volumes and at an affordable cost.

To date, the majority of studies that have attempted to identify molecular subgroups in OA have used blood samples (serum or plasma)[19–21]. The synovial fluid (SF), in contrast, offers a promising alternative discovery biofluid, as it is close to the diseased joint tissues and is enriched with locally derived biomolecules. Thus, SF is likely to represent more accurately the disease in a given joint. We have also previously shown that proteins in knee OA or after knee injury are readily detected in the SF but correlate poorly in paired blood[22–25]. Furthermore, we have confirmed the utility of large-scale protein measurements in SF using the SomaScan™ platform, an aptamer-based assay[26,27]. The SomaScan platform v4.1 measures 6596 distinct human proteins.

The Synovial fluid To Detect Endotypes by Unbiased Proteomics in OA (STEpUP OA) Consortium was established to test the primary hypothesis that there are detectable, distinct molecular endotypes in knee OA. We set out to perform an unsupervised analysis of a single SF sample from 1361 individuals with established OA, where cross-sectional clinical data were also available. The standardised protocol, which describes the cohorts in detail, and includes how we adjusted for pre-defined technical and other confounding factors is available elsewhere[27]. Here we present the primary analysis of STEpUP OA, in which we determine whether protein molecular endotypes exist in the SF of participants with established knee OA, and further explore the relationship between proteomic signatures and structural and symptomatic disease.

## RESULTS

### Endotype detection in OA SF
To search for molecular endotypes in OA using SF protein profiles, the f(K) cluster metric was employed. We had previously reported that a large contributor of variance in the initial processed data (principal component 1, accounting for 48% of variance), was due to intracellular proteins[27]. Appreciating that the intracellular protein signature could obscure subtle clustering patterns within the data, we performed cluster analyses with and without regression adjustment for intracellular protein[27], using an intracellular protein score (IPS) that correlated highly with principal component 1 ($r = 0.94$)[27]. Cluster analysis revealed 2 clusters that were evident within the Discovery, Replication and Combined datasets for the non-IPS regressed analysis (Fig. 1A, left panel). In contrast, no clusters were detected in the IPS-regressed

dataset (Fig. 1A, right panel). Visualisation of the proteomic data structure in two-dimensional space showed that the two clusters were indistinct and could be defined by dichotomising the continuous IPS, a feature that was lost after IPS regression (Fig. 1B).

Association testing of IPS with pre-defined clinical and technical features (N = 1134, spun OA samples only) demonstrated that IPS was significantly, but modestly, greater in females, greater in advanced radiographic disease, and was greater in SF samples with visual blood staining scores ≥2 (Table 1). We therefore repeated the cluster analysis, using IPS and non-IPS regressed datasets, but stratified by biological sex (Fig. 1C), radiographic disease severity (Fig. 1D), and presence of blood staining (Fig. 1E). As with our non-stratified analyses, clusters (again indistinct) were only identified in non-IPS regressed data. Collectively, these data suggest that there are two potential endotypes in the non-IPS regressed data, but they are on a continuum, defined by the IPS, and are not distinct. Furthermore, the cluster structure is independent of the stage of disease, biological sex, and visible blood staining.

### Synovial fluid protein associations with radiographic OA
We next examined which SF proteins were associated with radiographic disease severity. Over 1000 proteins were significantly associated with advanced radiographic disease severity (advanced (KL 3-4) vs. non-advanced (KL 0-2)) in each of the Discovery (N = 1021, 96.0% upregulated) and Replication datasets (N = 2524, 98.6% upregulated), with 688 (24.1%) proteins replicating across both datasets. Figure 2A shows the combined dataset where 3815 proteins were associated with radiographic disease severity. Top associated proteins that replicated (across Discovery and Replication cohorts) and that remained significant in the Combined dataset after cohort adjustment are labeled in orange. Protein abundance profiles for a selection of the labelled proteins were also significantly associated with ordinal KL grade, either significantly decreasing with worsening radiographic disease severity (LYVE1, IGFPB-6, FGFP1, sFRP-3) or increasing (TSG-6, sTREM-1, Activin A, RSPO2) (Fig. 2B). Two additional proteins, previously linked to OA, MMP-13[28] and COL2[29], followed this latter pattern. Using the Hallmark gene set repository, nine differentially expressed pathways were significantly enriched across at least one of the three datasets (Fig. 2C). Of these, "Epithelial Mesenchymal Transition (EMT)", "Complement" and "Angiogenesis" were significantly associated with advanced radiographic OA across all datasets. These remained significantly enriched in the Combined dataset after adjustment for haemoglobin A, a surrogate marker for blood in the SF[27]. Protein-protein interactions within each of the enriched pathways are shown in Fig. 2D-F. "EMT" contained a number of molecules previously associated with matrix remodelling in OA[30] including, but not limited to, TIMP1, TIMP3, MMP-2, TGFβ1 and VEGFA. The correlation between protein associations within the Discovery and Replication datasets was $r = 0.49$ ($p < 2.2e^{-16}$) (Fig. 2G). To address which tissues drive OA SF biology, a comparative analysis was performed using published RNAseq datasets of both OA cartilage and synovium compared with non-OA tissues[31,32]. Many strongly regulated SF proteins overlapped with gene regulation in solid tissues, and pathway analysis showed enrichment for EMT in both tissues, with the Complement and Coagulation pathways evident in synovium only (Supplementary Fig. 1A, B).

Correlation of corresponding protein effects before and after adjustment for cohort (as a random intercept) was high ($r = 0.88$, $p < 2.2e^{-16}$) (Supplementary Fig. 2A), irrespective of differences in radiographic disease severity across cohorts (Supplementary Fig. 2B). Pathway analysis showed a robust "EMT" signature, although "Complement" and "Angiogenesis" pathways were no longer significantly enriched across all datasets (Supplementary Fig. 2C). The volcano plot of proteins that were associated with radiographic disease severity, after adjustment for IPS, is shown in Supplementary Fig. 3A. Correlation of corresponding protein effects was also high ($r = 0.82$, $p < 2.2e^{-16}$) (Supplementary Fig. 3B) and pathway associations for "EMT",

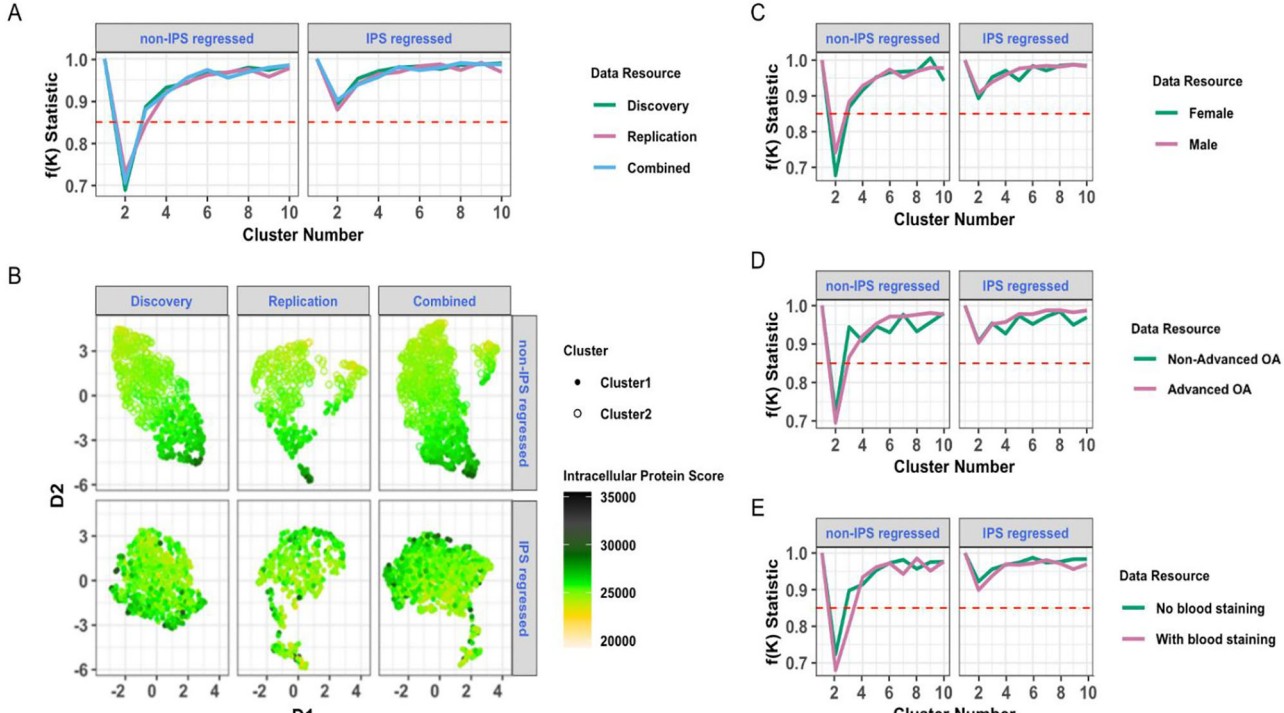

**Fig. 1 | Endotype discovery by cluster analysis in Discovery, Replication and Combined datasets. A** f(K) metric for non-IPS and IPS regressed analyses. Significant clustering was observed (f(K) < 0.85) across all three datasets (green = Discovery, pink = Replication, blue = Combined dataset) for non-IPS-regressed analyses only (left panel). **B** Visualisation of data structure and IPS on UMAP by dataset, stratified by non-IPS (top panel) and IPS regressed (bottom panel) analyses. f(K) metric plots for Combined dataset stratified by **C** biological sex (green = female, pink = male), **D** advanced radiographic status (KL grades: 0-2 as 'Non-advanced OA' (green) and ≥3 as 'Advanced OA' (pink)) or **E** blood staining (visual blood staining: 1 as 'No blood staining' (green) and ≥ 2 as 'With blood staining' (pink)) for non-IPS and IPS regressed analyses. OA osteoarthritis, IPS intracellular protein score, UMAP Uniform Manifold Approximation and Projection, KL Kellgren Lawrence.

"Complement" and "Angiogenesis" remained robust, but also included "Coagulation" (Supplementary Fig. 3C). Data associated with these analyses can be found in Source Data files 1 & 2. We also performed an analysis (not originally in the pre-published analysis plan) in which we compared the proteomes of knee SF from disease-free control participants (N = 36) with all knee OA cases (N = 1361). Over 1200 protein associations were observed (Supplementary Fig. 4A). Pathways identified by gene set enrichment analysis were similar to those identified in advanced vs. non-advanced disease, with significant correlation of associations between these analyses (r = 0.43, p < 2.2e⁻¹⁶) (Supplementary Fig. 4B, C, Source Data file 3).

**Synovial fluid protein associations with advanced radiographic OA after stratification by BMI or biological sex**

As "Metabolic OA", driven largely by BMI, has been suggested as a potential OA phenotype[33], we used STEpUP OA data to examine the proteins associated with radiographic disease severity after stratification by participant BMI (≥ 30 indicating obesity, N = 587, and <30, N = 649). We first looked at proteins in the SF that were associated with BMI, irrespective of radiographic disease status. Interestingly, a number of proteins known to be associated with BMI, including the appetite-suppressing hormone, leptin (LEP), insulin (INS), growth hormone receptor (GHR), and C-reactive protein (CRP), a well-validated inflammatory marker, were identified (N = 248, 66.9% upregulated) (Supplementary Fig. 5A; Source Data file 4). Leptin's SF levels correlated closely with BMI (r = 0.58, p < 2.2e⁻¹⁶) (Supplementary Fig. 5B), and associations of obesity-associated proteins appeared robust across datasets, and after cohort adjustment (Supplementary Fig. 5C–E). When stratified by obesity status, over 1800 proteins were significantly associated with advanced radiographic OA in each of the obese and non-obese groups (Fig. 3A, B), with a correlation between

the corresponding protein effects in the obese and non-obese groups of r = 0.72 (p < 2.2e⁻¹⁶) (Fig. 3C). No significant interaction terms with obesity status were identified by formal interaction testing (at padj <0.05). Interestingly, Hallmark pathway analysis showed a strong, consistent "EMT" pathway signature in both groups, but only samples from obese participants retained significant associations with "Coagulation" and "Complement" (Fig. 3D) (Source Data file 5). Consistent with their known associations with inflammation, complement and coagulation were also the pathways most strongly associated with CRP levels, even after adjustment for BMI (Supplementary Fig. 6A, E). CRP was significantly associated with both radiographic disease severity (logOR = 0.24, p-value = 0.00021) and WOMAC pain score (β = 1.83, p-value = 0.018)(Supplementary Fig. 6B, C). Protein associations with CRP are shown in Supplementary Fig. 6D and Source Data file 6.

To explore the influence of other participant factors on radiographic disease-protein associations, we also stratified samples by biological sex (Fig. 4A, B). Protein associations with radiographic disease severity, after stratification by biological sex, also had a strong correlation (r = 0.69, p < 2.2e⁻¹⁶, Fig. 4C), with 1437 significantly associated proteins common to the two groups. No significant interaction terms with biological sex were identified by formal interaction testing (at padj <0.05). Hallmark pathway analysis showed a strong "EMT" pathway signature in both sexes, but only males showed significant associations with "Angiogenesis" and "Coagulation" (Fig. 4D) (Source Data file 7), both of which remained significant after adjustment for haemoglobin A (Fig. 4D).

**Synovial fluid protein associations with WOMAC pain in OA**

Finally, we explored the association of SF proteins with patient-reported pain. We identified 797 (73.0% upregulated) SF proteins that were significantly associated with WOMAC knee pain in the Combined

**Table 1 | Baseline characteristics of participants, their SF samples and association of these factors with IPS**

| Feature | Description | Spun OA Samples (N) | Mean (SD) or N | Reference Group | Strength and Direction of Association of IPS (regression coefficient) Cohort included as a random intercept | | Adjusted p-values Cohort included as a random intercept | |
|---|---|---|---|---|---|---|---|---|
| | | | | | Yes | No | Yes | No |
| **Age** | Participant age at the time of sampling (year) | 1133 | 64.46 (11.00) | — | $2.01e^{-05}$ | $1.47e^{-04}$ | $7.01e^{-01}$ | $2.64e^{-01}$ |
| **Sex** | Biological sex | 1134 | Female: N = 596 Male: N = 538 | Female | $-5.47e^{-03}$ | $-6.29e^{-03}$ | $2.42e^{-02}$* | $1.22e^{-02}$* |
| **BMI** | Participant body mass index at the time of sampling | 1045 | 30.68 (5.92) | — | $4.11e^{-05}$ | $7.87e^{-05}$ | $8.09e^{-01}$ | $7.59e^{-01}$ |
| **Smoking History** | Current or past smoker at the time of the baseline sampling | 926 | Never Smoked: N = 510 Ever Smoked: N = 416 | Never Smoked | $1.38e^{-03}$ | $7.26e^{-04}$ | $7.01e^{-01}$ | $7.59e^{-01}$ |
| **WOMAC Pain Score** | Scale of 0-100, where 100 is the worst possible knee pain | 748 | 44.91 (21.08) | — | $2.63e^{-05}$ | $3.81e^{-05}$ | $7.01e^{-01}$ | $7.59e^{-01}$ |
| **Advanced Radiographic Status** | Binary indicator for the presence of advanced radiographic knee OA (KL grades 3-4) | 1096 | Non-Advanced: N = 264 Advanced: N = 832 | Non-Advanced OA (KL grades 0-2) | $1.07e^{-02}$ | $1.09e^{-02}$ | $4.36e^{-04}$* | $2.18e^{-04}$* |
| **Visual Blood Staining Grade** | Grading of SF blood staining (BS) prior to centrifugation (if known). Scale of 1-4, with larger grades corresponding to greater degrees of blood staining | 515 | Grade 1: N = 394 Grade 2: N = 77 Grade 3: N = 26 Grade 4: N = 18 | BS = 1 | BS = 2, $1.50e^{-02}$ BS = 3, $3.48e^{-02}$ BS = 4, $5.54e^{-02}$ | BS = 2, $1.59e^{-02}$ BS = 3, $3.60e^{-02}$ BS = 4, $5.68e^{-02}$ | $8.53e^{-13}$* | $6.51e^{-13}$* |

Association testing was carried out between IPS and core demographic, clinical and technical features in spun OA samples where relevant data were available. Linear regression models were constructed with log scaled IPS (i.e. IPS that was transformed using natural logarithms) as the outcome with each feature listed in the table used as a univariate exposure. Adjusted models where cohort was included as a random intercept are also shown. Asterisks (bold) denote statistical significance at Benjamini-Hochberg cutoff (adjusted p-value ≤ 0.05).

OA osteoarthritis, SF synovial fluid, IPS intracellular protein score, BS blood staining, KL Kellgren Lawrence, SD standard deviation, BMI body mass index, WOMAC Western Ontario and McMaster Universities Osteoarthritis Index, 'advanced' (KL grades: 3-4), 'non-advanced' (KL grades: 0-2).

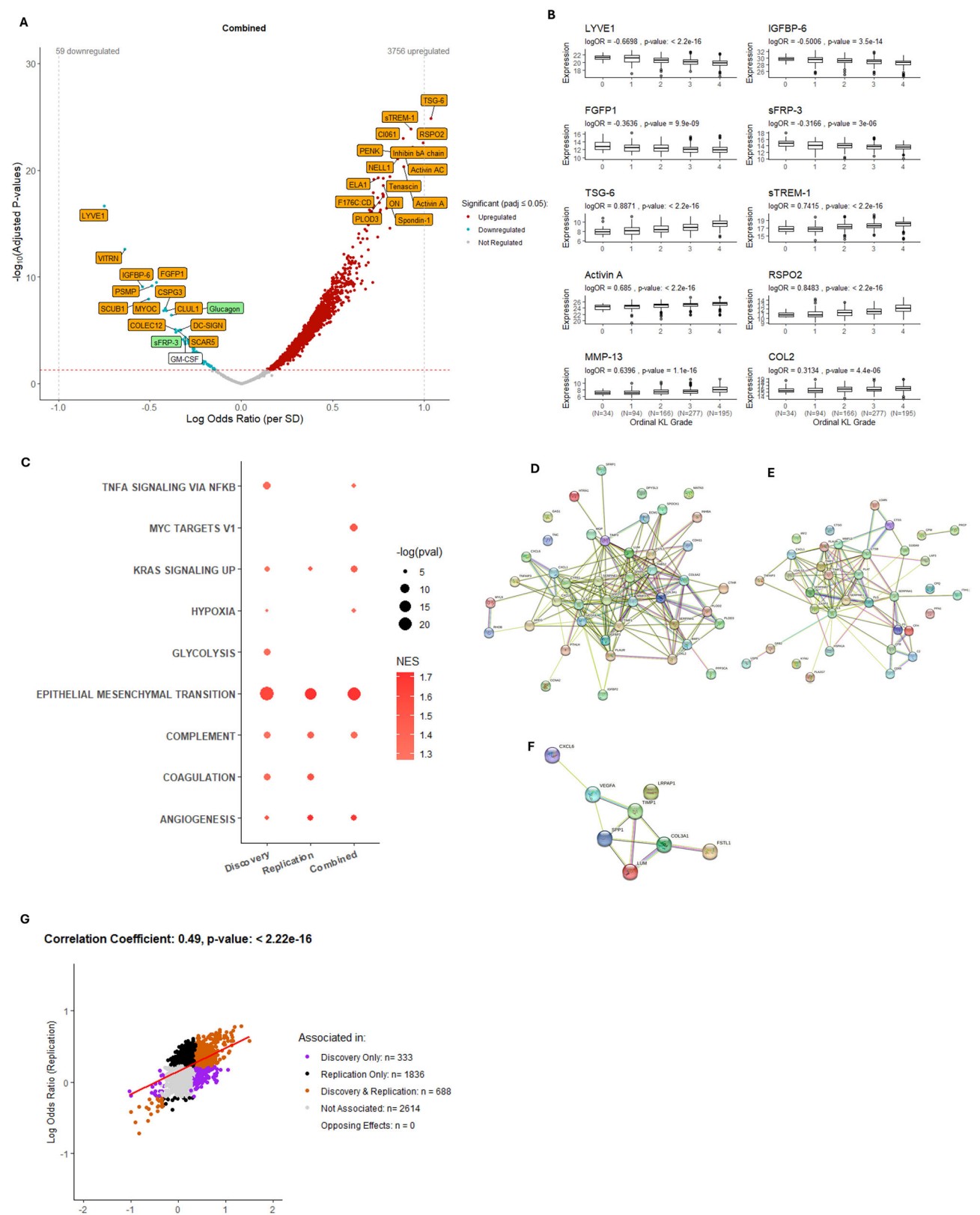

non-IPS regressed dataset. However, none of these proteins replicated across Discovery and Replication datasets and the cross-dataset correlation was 0.36 (p < 2.2e⁻¹⁶) (Fig. 5A, B). Noelin-2 (NOE2) and ecto-ADP-ribosyltransferase 3 (NAR3) were the only significantly associated proteins in the Combined dataset after cohort adjustment (Supplementary Fig. 7A and labelled green in Fig. 5A) (Source Data file 8). The

associations between NOE2 and NAR3 protein abundance with WOMAC pain subscores are shown in Fig. 5C (linear regression). The pathway analysis did not identify consistent associations across Discovery, Replication and Combined datasets (Fig. 5D). WOMAC knee pain subscores were unevenly distributed across Discovery and Replication cohorts (Supplementary Fig. 7B). The number of proteins

**Fig. 2 | Association between protein abundance and advanced radiographic knee OA status in non-IPS regressed data using logistic regression modelling.** Protein abundance was measured in 1,322 samples (Combined: 1,096 spun, 226 unspun), adjusted for spin-status (using ComBat) and then age and biological sex. This included 1,016 advanced and 306 non-advanced cases. **A** Volcano plot: logORs for proteins associated with advanced radiographic status (KL grades 3-4) in Combined dataset against Benjamini-Hochberg adjusted p-values (padj). Positively (red) and negatively (blue) associated proteins are shown (padj ≤ 0.05). Top 30 proteins, by padj, are labelled. Proteins that replicated (significant (padj ≤0.05) with consistent effects in both Discovery & Replication) and remained significant after cohort adjustment in Combined dataset are orange. Proteins that either did not replicate but remained significant after cohort adjustment, or replicated but lost significance after cohort adjustment, are green. **B** Boxplots: log-transformed protein expression by ordinal KL grade (Combined: N = 766). Associations with KL grade were tested by proportional odds ordinal regression, adjusted for age and biological sex. LogOR and unadjusted p-values are shown, with sample counts per KL grade. Two additional OA-related proteins (MMP-13 & COL2) are included.

Boxplots show median, interquartile range, and whiskers representing the most extreme values within 1.5 times the interquartile range, with outliers plotted individually. **C** Bubble plot: enriched pathways (padj <0.05) for advanced radiographic disease across datasets. **D–F** Protein-protein interaction networks for Epithelial-Mesenchymal Transition, Complement, and Angiogenesis pathways, built using the top 1,000 proteins (by logOR). **G** Scatter plot: logOR from logistic regression models of protein abundance with advanced radiographic disease status shows significantly associated proteins (padj ≤ 0.05) in the Discovery and Replication datasets. Pearson correlation coefficient and p-value (unadjusted) are presented. IPS intracellular protein score, KL Kellgren-Lawrence, logOR log odds ratio, padj adjusted *p*-value, LYVE1 Lymphatic vessel endothelial hyaluronan receptor 1, IGFBP-6 Insulin-like growth factor-binding protein-6, FGFP1 Fibroblast Growth Factor Binding Protein-1, sFRP-3 secreted frizzled-related protein 3, TSG-6 tumour necrosis factor-inducible gene 6, sTREM-1 soluble triggering receptor expressed on myeloid cells-1, RSPO2 R-spondin-2, MMP-13 Matrix metalloproteinase-13 and COL2 Collagen Type II, GM-CSF Granulocyte-macrophage colony-stimulating factor, NES normalised enrichment score. Data available in Source Data file 1.

associated with pain was also reduced in the Combined dataset after adjustment for radiographic disease severity (Supplementary Fig. 7C, Source Data file 9). NOE2 and NAR3 remained significantly associated with WOMAC pain after adjustment, and their levels were not independently associated with radiographic grade (by ordinal regression) (Supplementary Fig. 7D). The correlation between pain-associated protein effects from non-IPS and IPS regressed analyses using the Combined datasets was r = 0.97 (p < 2.2e$^{-16}$) (Supplementary Fig. 7E). Interestingly, nerve growth factor (NGF), the best validated pain target in OA[34–36], was associated with increased radiographic disease severity (Combined dataset, logOR = 0.269, padj = 0.002), but not with WOMAC knee pain (β = 1.157, padj = 0.40). The top 20 proteins associated with each of the clinical outcomes (by padj) compared with the logORs for all OA versus disease-free controls (for that given protein) are shown in Supplementary Table 1.

## Discussion

We describe here the primary results of STEpUP OA, the largest, unbiased, replicated, cross-sectional synovial fluid proteomic analysis of knee OA to date. We uncover the balance of biological pathways in disease and how they change with structural and symptomatic disease severity, as well as by important patient-related factors, such as obesity and biological sex. The data presented here do not reveal evidence for distinct molecular endotypes. Rather, they indicate that OA is a biological continuum, with individuals distributed along a spectrum for a given biological pathway. Such information is likely to be helpful in selecting the right therapy for the right individual.

Synovial fluid is an ultrafiltrate of the plasma but also reflects joint-specific processes such as active secretion from cells[37], including extracellular vesicles, release from damaged or short-lived cells, and shedding from cell and tissue surfaces. Pathway analysis of OA SF proteins associated with radiographic disease severity indicated a robust activation of "EMT", indicative of active tissue remodelling, presumably part of the joint injury response[38]. This was also evident in both OA cartilage and synovium RNAseq analyses, which showed enrichment for the EMT pathway. Thus, the cartilage as well as the synovium contributes to SF biology, as others have suggested[31,37,39]. Of note, complement and coagulation pathways were only enriched in the synovium. These pathways varied by patient stratification (BMI, sex) and with CRP level. Successful therapeutic targeting has been demonstrated in murine OA for both complement and coagulation, suggesting that their levels in SF may help stratify individuals who could benefit from such targeting[40–42]. These results are consistent with OA synovium histology which shows a continuum of tissue hyperplasia and modest inflammatory cell infiltration[43], and which is quite distinct from pathotypes described in rheumatoid arthritis[44]. The

lack of a strong immune signature is consistent with OA genome-wide association studies[45] and the data we present in this manuscript.

Replication in STEpUP OA was robust for associations with structural disease but less so for pain. This lack of association is unlikely to be because the joint does not contain molecules that are directly involved in triggering pain responses, as most individuals (>80%) gain symptomatic benefit after joint replacement. It seems more likely that this is due to high variability in patient-reported symptomatic outcomes, which are known to be influenced by external factors beyond molecular drivers made by the joint, e.g., psychological factors[46], biological sex. This makes cross-sectional analyses of this sort challenging. Levels of pain will also be influenced by analgesic treatments that the patient was taking, although this was not captured comprehensively across the whole cohort and was not, therefore, included as a potential confounding factor. Only a small proportion of the individuals in STEpUP OA had associated prospective clinical outcome data (mainly pain scores). These were not included in the current reported study but will be examined separately in future work. Protein associations with pain may also have been limited by the fact that WOMAC pain scores were only available on a subset within STEpUP OA (N = 805) and most of these were within a relatively narrow range of pain severity. Whilst protein associations with pain lacked replication, there were, nonetheless, a number of significantly regulated molecules of interest identified in the combined analysis, including noelin-2, a component of the AMPA glutamate receptor and involved in muscle differentiation[47]. Further validation of these associations is required.

Despite this being the largest analysis of its kind, we recognise a number of limitations: firstly, protein detection using the SomaScan platform for SF is still relatively new and it is possible that the method and/or SF might not be optimal to disclose endotypes. It is reassuring, in this regard, that molecular endotypes have been discovered in asthma, using SomaScan in both serum and induced sputum samples[48,49]. Our samples were generated from a diverse set of, largely, pre-existing cohorts. The percentage who had successful SF aspiration was documented in only 4/17 of the cohorts (albeit accounting for 51% of the total participant number). In these instances, successful aspiration of SF was greater than 65%, but it remains unclear how representative this is of the whole cohort and whether this may have biased the biology revealed in our analysis and the generalisability of OA. Our analysis was powered to identify several endotypes across the entire OA population and to detect two distinct endotypes when considering only non-advanced radiographic disease.

The cross-sectional analysis presented in this manuscript provides strong proof of concept that knee OA synovial fluid provides an informative window into disease-relevant biology. Discernible patient

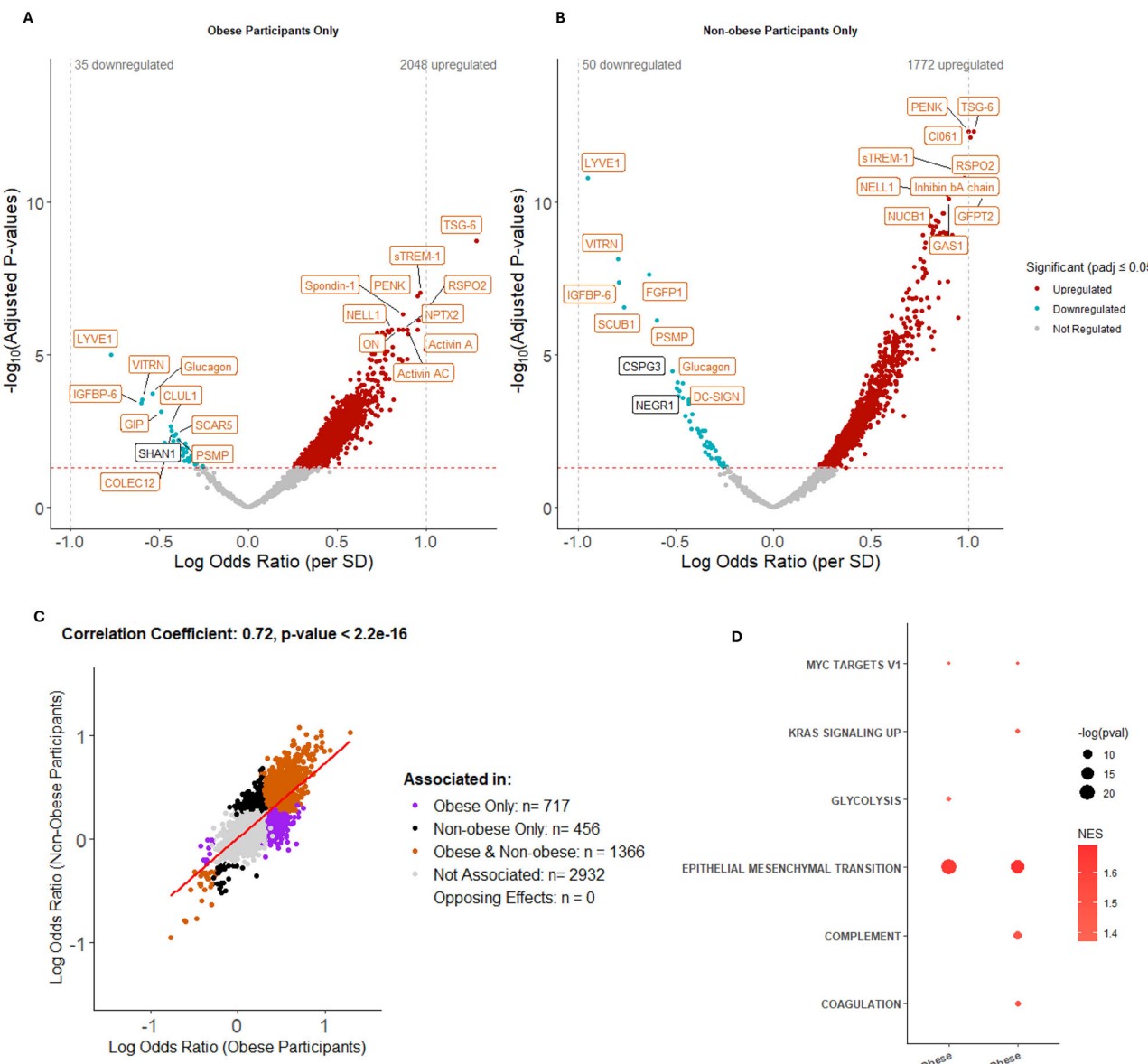

**Fig. 3 | Association between protein abundance and advanced radiographic disease status stratified by obese and non-obese OA participants in non-IPS regressed Combined data using logistic regression modelling.** Protein abundance was measured in 1236 samples where BMI was available (Combined: 1045 spun, 191 unspun), adjusted for spin-status (using ComBat) and then age and biological sex. The groups were then stratified by BMI into obese, BMI ≥ 30 (N = 587, 504 spun samples) and non-obese, BMI < 30 (N = 649, 541 spun samples) participants. Volcano plot showing log odds ratios (logOR) per standard deviation change in protein expression for proteins associated with advanced radiographic status (KL grades 3-4) in the Combined dataset, with Benjamini-Hochberg adjusted p-values (padj), in **A** obese and **B** non-obese groups. Proteins in red are positively associated, and those in blue are negatively associated with advanced radiographic status (padj ≤ 0.05). Top 20 associated proteins in each direction, by padj, are labelled. In orange are proteins that replicated (significant at padj ≤0.05 and with effects in the same direction) in obese and non-obese groups, whereas white labelled proteins were only associated in the obese-status specific set. **C** Scatter plot of logOR from logistic regression models of the associations between protein abundance and advanced radiographic disease status in obese and non-obese groups is shown, with significantly associated proteins (at padj ≤ 0.05) in different groups shown in different colours (see key). Pearson correlation coefficient and p-value (unadjusted) are presented for the correlation between logOR generated in obese only and non-obese-only analyses (Combined dataset). **D** Bubble plot of significantly enriched pathways (padj <0.05) using the Hallmark Gene set for proteins associated with advanced radiographic disease status by obesity status. IPS intracellular protein score, logOR log odds ratio, SD standard deviation, padj adjusted *p*-value, NES normalised enrichment score. The full list of proteins is available in Source Data file 5.

molecular clusters from OA relevant tissues, have been described in OA cartilage and synovium[31,50–52], in SF using mass spectrometry[53,54], and in plasma using candidate biomarkers[17,19,21]. However, these studies are considerably smaller than STEpUP OA, included only a few replications, and, where identified, clusters were continuous rather than distinct. Several examined prospective outcomes associated with clusters, rather than the cross-sectional analysis that we present here.

Prospective analyses in a subset with longitudinal data are now planned in STEpUP OA. Future studies will also include a multi-omic approach using data from paired genetic and metabolomic analyses. Ultimately, we hope that SF analyses of this sort will assist in stratifying individuals to enrich recruitment into experimental medicine studies to de-risk subsequent clinical trials. The publication of this manuscript also marks the opportunity to welcome external parties to apply for

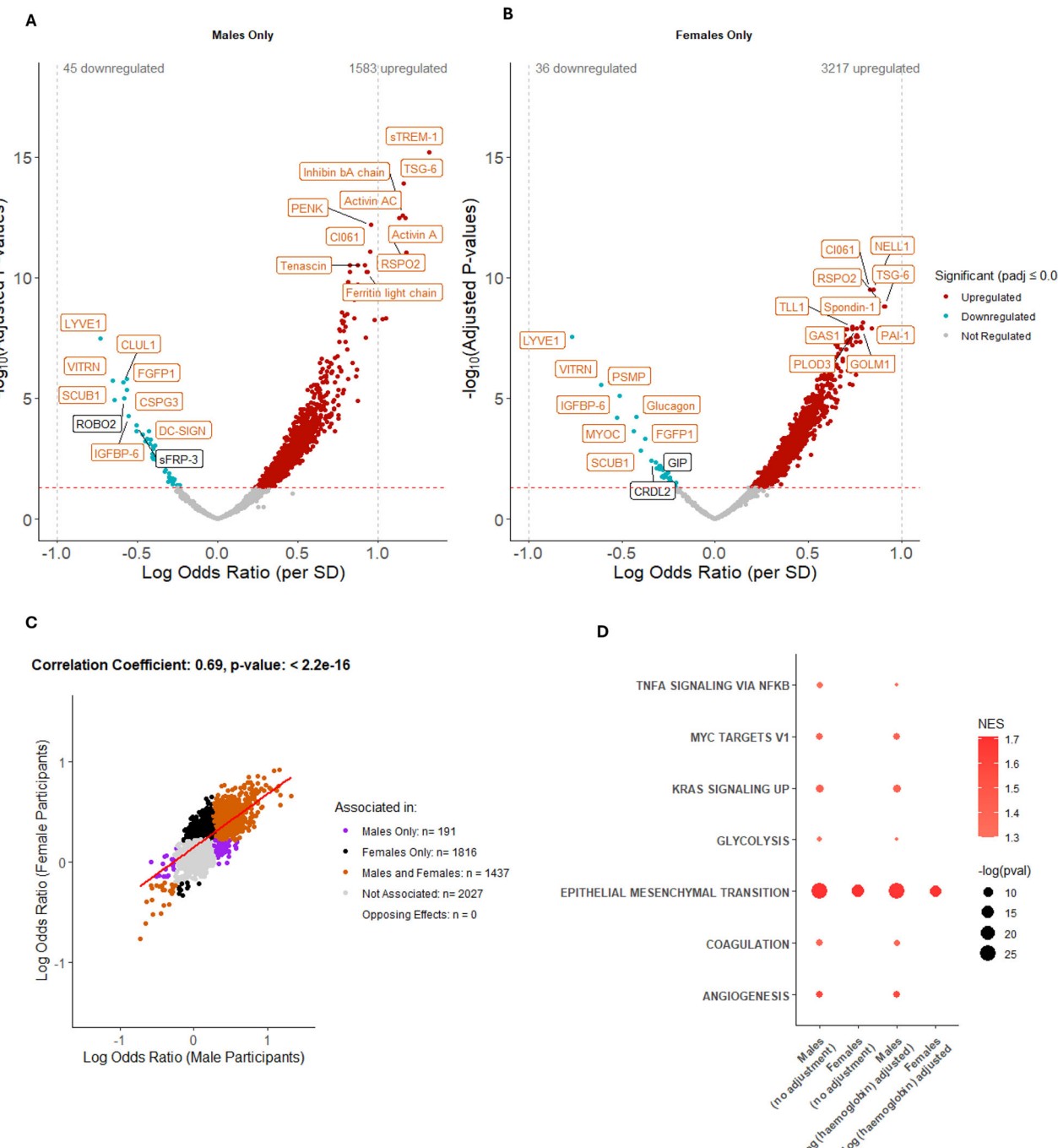

**Fig. 4 | Association between protein abundance and radiographic OA severity after stratifying for biological sex in non-IPS regressed Combined data using logistic regression modelling.** Protein abundance was measured in 1322 samples (Combined: 1096 spun, 226 unspun), adjusted for spin-status (using ComBat) and then age. Volcano plot showing log odds ratios (logOR) per standard deviation change in protein expression for proteins associated with advanced radiographic status (KL grades 3-4) in the Combined dataset, with Benjamini-Hochberg adjusted p-values (padj), in **A** males (N = 623) and **B** females (N = 699). Proteins in red are positively associated, and those in blue negatively associated with advanced radiographic status (padj ≤ 0.05). Top 20 associated proteins in each direction, by padj, are labelled. In orange are proteins that replicated (significant at padj ≤0.05 and with effects in the same direction) in males and females, whereas white labelled proteins were only associated in the sex-specific set. **C** Scatter plot of logOR from logistic regression models of the associations between protein abundance and advanced radiographic disease status in males and females is shown with significantly associated proteins (at padj ≤ 0.05) in different groups shown in different colours (see key). Pearson correlation coefficient and p-value (unadjusted) are presented for the correlation between logOR generated in male-only and female-only analyses (Combined dataset). **D** Bubble plot of significantly enriched pathways (padj <0.05) using the Hallmark Gene set for proteins associated with advanced radiographic disease status by biological sex with and without additional adjustment for log (haemoglobin A protein expression). IPS intracellular protein score, SD standard deviation, logOR log odds ratio, padj adjusted *p*-value, SD standard deviation, NES normalised enrichment score. The full list of proteins is available in the Source Data file 7.

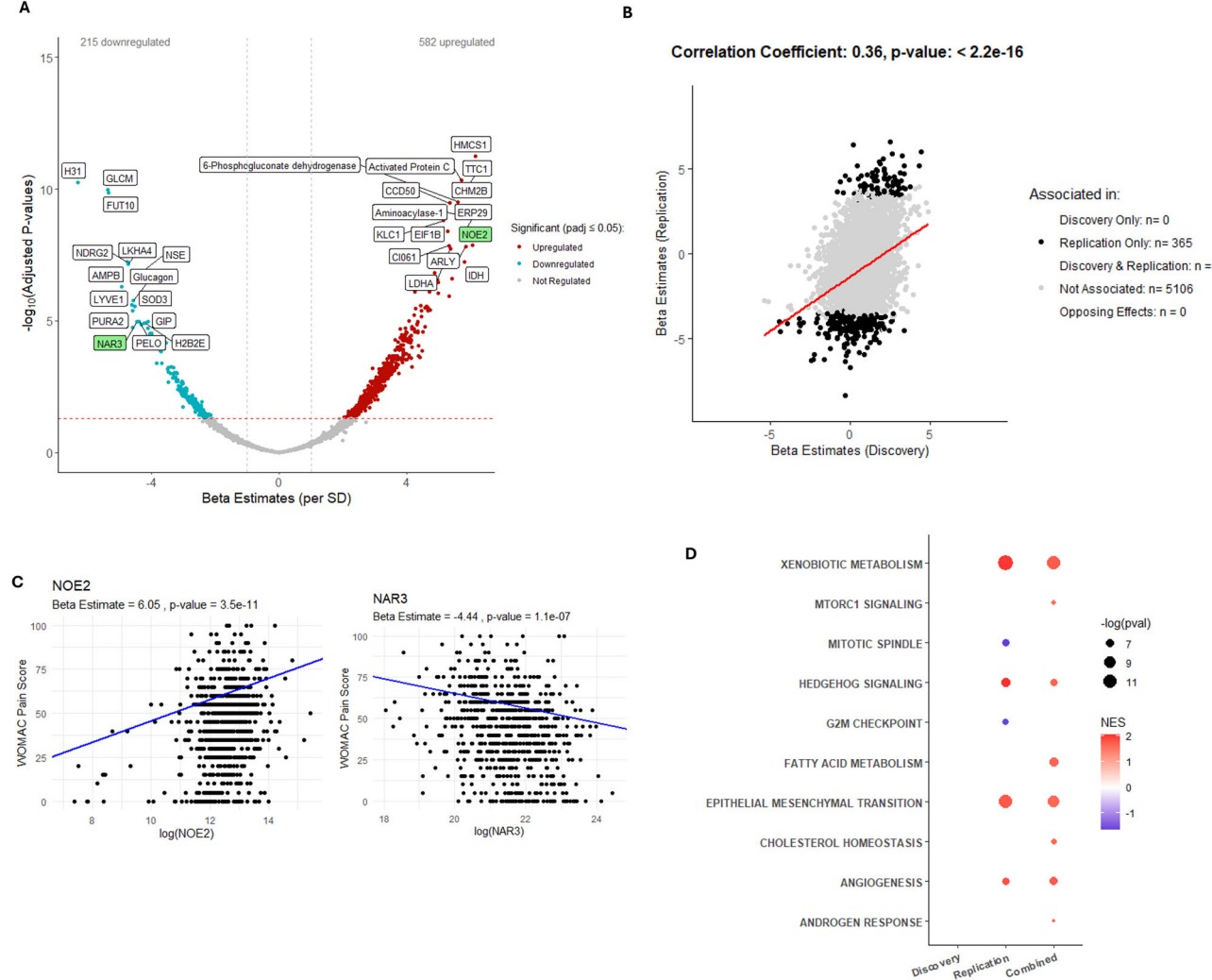

**Fig. 5 | Association between protein abundance and WOMAC knee pain subscore in non-IPS regressed data using linear regression modelling.** Protein abundance was measured in 805 OA samples where WOMAC knee pain subscore data was available (Combined: 748 spun, 57 unspun), adjusted for spin-status (using ComBat) and then age and biological sex. **A** Volcano plot showing beta estimates per standard deviation change in protein expression for proteins associated with WOMAC knee pain subscore in the Combined dataset, with Benjamini-Hochberg adjusted p-values (padj). Proteins in red are positively associated, those in blue are negatively associated, with increasing WOMAC knee pain subscore at padj ≤ 0.05. Top 30 associated proteins, for each direction, ordered by padj are labelled. In green are two proteins that were significant in Replication and Combined datasets (at padj ≤ 0.05), including after cohort adjustment, see Supplementary Fig. 7. **B** A scatter plot of beta estimates from linear regression models of the associations between protein abundance and WOMAC knee pain in non-IPS analyses is shown for Discovery and Replication datasets with significantly associated proteins (at padj ≤ 0.05) in different groups shown in different colours (see key). Pearson correlation coefficient and p-value (unadjusted) are presented for the correlation between beta estimates generated in non-IPS regressed analyses using Discovery and Replication datasets. **C** Scatter plots of WOMAC pain subscore against NOE2 or NAR3 protein abundance (transformed by natural logarithms) in OA participants using Combined, spin-status corrected, non-IPS regressed data. Beta estimates and p-values (unadjusted) are presented for linear models adjusted for age and biological sex. **D** Bubble plot of significantly enriched pathways (padj <0.05) using the Hallmark Gene set for proteins associated with WOMAC knee pain subscore by Replication and Combined datatsets not adjusted for IPS or cohort. No pathways were significantly enriched at padj <0.05 in the Discovery dataset. OA osteoarthritis, SD standard deviation, IPS intracellular protein score, WOMAC Western Ontario and McMaster Universities Osteoarthritis Index (WOMAC, 0 = no pain, 100 = worst possible pain), noelin-2 (NOE2); ecto-ADP-ribosyltransferase 3 (NAR3), (padj) adjusted p-value, NES normalised enrichment score. Full list of proteins available in Source Data file 8.

access to STEpUP OA data for research purposes in accordance with our Consortium Agreement.

## METHODS

### Study design principles

STEpUP OA was set up to search for molecular endotypes in knee OA. The primary analysis of STEpUP OA utilised data and samples from 17 cohorts, where an SF sample was available (N = 1361 participants meeting consortium eligibility criteria for knee OA; N = 36 control samples (disease-free participants))[27]. All participants gave written informed consent with local ethical approvals in place. The University of Oxford Medical Sciences Central University Research Ethics Committee (CUREC) granted ethical approval for the processing, storage and use of samples and linked data for STEpUP OA (R67029/RE001). Our study abides by the declaration of Helsinki. Individual cohorts were assigned, a priori, into Discovery (N = 708) and Replication (N = 653) datasets (Supplementary Table 2). Most samples were centrifuged ('spun') after joint aspiration but appropriate correction was applied for non-centrifuged ('unspun') samples. Full details of the cohorts and their associated metadata, how SF was collected and processed, and how we corrected for pre-defined technical and other confounders can be found elsewhere[27]. SF sample numbers and

number of SOMAmers[TM][27] for each experiment varied according to data availability, adjustments made, and analysis performed (Supplementary Table 3).

## Analysis platform

All SF samples were analysed on the SomaScan platform v4.1 (SomaLogic); a high-throughput, aptamer-based proteomics assay designed for the simultaneous assessment of 7596 synthetic DNA slow off-rate modified aptamers (SOMAmers) (7289 unique human targets)[27]. SF samples were randomized and analysed as a single batch at SomaLogic (Boulder, CO, USA). Following filtering for poor performing SOMAmers[TM][27], our analyses included protein data from between 5278 and 6558 SOMAmers (Supplementary Table 3).

## Statistical analysis

**Quality control of proteomic data.** All proteomic data received from SomaLogic underwent pre-processing and quality control procedures as previously reported[27]. Briefly, raw data was standardised using a modified version of SomaLogic's normalization pipeline and batch-effect correction, followed by removal of samples and aptamers of insufficient quality, to produce our initial downstream dataset for future analyses. All statistical analyses were pre-specified and outlined in our data analysis plans (see below).

**Unsupervised clustering for endotype detection.** Dimension reduction on batch-corrected, log-transformed proteomic data was performed using unscaled Principal Component Analysis (PCA), with the top principal components explaining 80% variation. Unsupervised clustering was performed in the reduced feature space using k-means with 10 random initializations. We tested for the presence of significant clusters using the f(K) statistic[55]; with the f(K) statistic visualised across cluster numbers. Data were determined to be significantly clustered if, for any number of clusters K, f(K) < 0.85 (a priori specified). Elbow plots were constructed to test the robustness of our findings. If the data were significantly clustered, we picked the optimal cluster number by majority vote across different clustering metrics (as implemented in the R package *NbClust*[56], version: 3.0.1) for downstream analyses. Clustering structure was visualised using Principal Component (PC) and Uniform Manifold Approximation and Projection for Dimension Reduction (UMAP)[57] plots.

**Protein–clinical feature association testing.** Associations between protein expression and clinical outcomes were modelled by fitting regression models for each SOMAmer separately, with clinical features set as the dependent variable and log-expression for each protein set as the independent variable. Linear, logistic, or proportional odds ordinal regression models were fitted for continuous, binary, or ordered categorical variable outcomes, respectively. Residual diagnostics confirmed adequacy of model assumptions. Before fitting the models, protein expression values were transformed using natural logarithms and were standardized on a per protein basis (within Discovery, Replication, and Combined datasets) by subtracting the mean log protein abundance and then dividing by its standard deviation, to make the slopes comparable between models. The resulting beta estimates (β, from linear regression models) or log odds ratios (logOR, from logistic and ordinal models) can be interpreted as either the mean outcome change or the logOR per standard deviation change in the log protein abundance. Replication was defined as proteins that were significant at Benjamini-Hochberg[58] adjusted p-value (padj) ≤ 0.05 (with no fold change thresholds set) in both Discovery and Replication datasets and with effects in the same direction.

The primary regression models (non-stratified) were adjusted for age and biological sex. All analyses were batch corrected for spin-status (using the R function *ComBat*[59,60], version 0.0.4) and run in duplicate using either proteomic data that had undergone further

regression adjustment for intracellular protein score (IPS)[27] ('IPS regressed') or without ('non-IPS regressed'). Association testing between IPS, which had been transformed using natural logarithms, and demographic, clinical, and technical features was performed using regression modeling, with all analyses either unadjusted or adjusted for cohort (as a random intercept). Volcano plots were generated to display associated proteins from the regression analyses, with the most strongly positively and negatively associated proteins (by padj) labelled by SOMAmer protein target name. A small number of proteins (between N = 375 and N = 383, according to correction) had more than one detection SOMAmer on the platform. Where this was the case, only the most significant (by padj) SOMAmer was labelled on the volcano plot. We also conducted interaction testing for associations between protein abundance and clinical features of disease (advanced radiographic status (Kellgren Lawrence [KL] grade ≥3) and WOMAC knee pain[61] (transformed to a scale of 0-100, 100 = worse possible pain)). A protein abundance-by-biological sex interaction term was included to test explicitly whether biological sex modified the association between protein abundance and the given outcome. Similarly, a protein abundance-by-obesity status (a dichotomous variable, BMI ≥ 30 kg/m$^2$) interaction term was examined. Pre-specified clinical outcomes used in association testing are listed in Table 1 & Supplementary Table 2. All other adjustments are described in Supplementary Table 3.

**Pathway enrichment analysis.** We tested for enrichment of associated proteins within pathways using gene sets taken from The Molecular Signatures Database (MSigDB, https://www.gsea-msigdb.org/gsea/msigdb); specifically, Hallmark, Gene Ontology (GO), Reactome, and Kyoto Encyclopedia of Genes and Genomes (KEGG)[62,63]. All proteins were mapped to the corresponding gene set based on 'EntrezGeneSymbol', 'Target' or 'EntrezGeneID' variables provided by SomaLogic. Protein set enrichment testing was performed using the *fgsea*[64] package in R (version: 1.28.0) to identify pathways whose genes were enriched for association with a given outcome. All proteins featured in the respective regression models were ranked by a 'rank metric' calculated as; rank metric = -log(p-values) * sign(β or log logOR per standard deviation change in protein expression). The sign function returns +1 if the estimate is positive, -1 if it is negative, and 0 if it is zero, thereby capturing the direction of effect. Enrichment scores were calculated as the maximum value of the running sum and normalized relative to pathway size, resulting in Normalized Enrichment Scores (NES). The direction and magnitude of pathway enrichment for a given outcome (i.e. differential regulation of the pathway) was determined using NES. The *ggplot2*[65] R package (version: 3.5.0) was used to draw bubble plots and visualise results.

Protein-protein interaction (PPI) networks were constructed using the top 50-1000 proteins (by absolute β or logOR), using the Search Tool for the Retrieval of Interacting Genes/Proteins database (STRING version 11.5, https://string-db.org/[66]). The filter condition was set as follows: network type selected; full-STRING network; confidence ≥0.2–0.4.

**Comparisons with published RNA Sequencing Data.** Published RNA sequencing gene expression data were analysed; one study comparing OA vs. non-OA cartilage (N = 44 OA cases (total knee replacement) and N = 10 non-OA controls)[31] and a second study comparing OA vs. healthy synovium (GSE89408)[32]. For cartilage, RNA-Seq summary statistics including log$_2$ fold change values, p-values, and adjusted p-values were examined. To compare these data with SF protein associations with advanced radiographic disease from STEpUP OA, we created a dataset mapping gene names between the RNA-Seq and STEpUP OA datasets. The RNA-Seq dataset contained 60,808 genes, while the STEpUP OA dataset included 5,471 proteins. Duplicates were removed, where present in either dataset,

by selecting the gene/protein with the smallest adjusted *p*-value, resulting in 4,907 proteins and 58,821 genes. Mapping by gene name produced a paired dataset of 4,832 gene-protein pairs. Pathway enrichment analysis was carried out on the full RNA-Seq dataset (N = 48,428 genes with available *p*-values) using the same approach as for proteins described above.

For synovium, an RNA-Seq count matrix for synovial biopsies (N = 28 normal, N = 22 OA) was extracted from (GSE89408)[32]. We performed differential expression testing between OA and normal samples using *DESeq2* (version 1.42.1) using default parameters and settings, generating log$_2$ fold change values, *p*-values, and adjusted p-values for 25,022 genes. STEpUP OA proteins were mapped to genes, forming a paired dataset of 4,364 gene-protein pairs. Pathway enrichment analysis was performed on the full RNA-Seq dataset (N = 25,022 genes with available *p*-values).

**Statistical significance.** Pearson correlation coefficient and relevant p-values are given for both correlation testing and regression modelling. All analyses were carried out in R Statistical Software (v4.3.2; R Core Team 2023)[67]Statistical significance was defined using Benjamini-Hochberg corrected p-values adjusted for multiple testing (padj), at a false discovery rate (FDR) of 5%.

## Data analysis plan
https://www.kennedy.ox.ac.uk/oacentre/stepup-oa.

## Data availability
SomaScan data of all healthy and OA synovial fluid (fully quality controlled as per Deng et al. 2024) are available in Figshare (https://doi.org/10.6084/m9.figshare.31626121). All code used to generate the tables and figures in this manuscript are provided here, GitHub. Participants' written informed consent provided for this study prevents unrestricted public sharing of individual-level research data except by collaboration. Access to the pseudonymised individual participant data supporting this work is available upon completion of a STEpUP OA Data Access Request which should be emailed to: stepupoa@kennedy.ox.ac.uk. Data reuse, publication and authorship requirements are indicated on the Access Request form. Response time will be within 12 weeks. Where possible, source data are provided with this paper (source data: 1-9). Source data are provided with this paper.

## Code availability
All R code, including the html vignette, are available at https://github.com/ndorms-tperry/STEpUP-OA-Primary-Manuscript[68].

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

## Acknowledgements

We would like to express our gratitude and thanks to all cohort participants who contributed samples to STEpUP OA. We are grateful for the support from Floris Lafeber and Simon Mastbergen (Utrecht Medical Centre) for provision of samples. We thank the Oxford Knee Surgery Team. We thank Gretchen Brewer for her administrative support of the

consortium. We thank Dr Jamie Soul (University of Liverpool) for his assistance in providing raw cartilage RNA sequencing data. The study was supported by Kennedy Trust for Rheumatology Research (grant number: 171806), Versus Arthritis (grant number: 22473), Centre for Osteoarthritis Pathogenesis Versus Arthritis (grant numbers: 21621, 20205), Galapagos, Biosplice, Novartis, Fidia, UCB, Pfizer (non-consortium member) and Somalogic (in kind contributions). The funders Kennedy Trust for Rheumatology Research, Versus Arthritis and Pfizer had no role in the study design, data collection and analysis, decision to publish or preparation of the manuscript. The funders Galapagos, Biosplice, Novartis, Fidia, UCB and SomaLogic were all active consortium members, attending consortium meetings. As such they made contributions to the study design and support of data collection, decision to publish and review and commenting on the manuscript. In addition, SomaLogic (now known as Standard BioTools), UCB and Novartis were members of the Data Analysis Group.Additional relevant funding sources: LJD is supported by a Wellcome Trust fellowship grant 208750/Z/17/Z and Kennedy Trust for Rheumatology Research for the present manuscript. FEW was directly supported in this work by her UKRI Future Leaders Fellowship and its renewal (MR/S016538/1;MR/S016538/2; MR/Y003470/1). FW, NKA and SK are members of the Centre for Sport, Exercise and Osteoarthritis Research Versus Arthritis (grant number 21595). MK is supported by grants from CIHR, NSERC, The Arthritis Society Canada, Krembil Foundation, CFI, Canada Research Chairs program, and has received support from the University Health Network Foundation, Toronto for the present manuscript. TJW is supported by grants from NWO-TTW Perspectief (#P15-23), Stichting de Weijerhorst and ReumaNederland (LLP14) for the present manuscript. CTA is supported by the Canadian Institutes of Health Research, Western University Bone and Joint Institute, and the Academic Medical Organization of Southwestern Ontario for the present manuscript. BDMT is supported through the United Kingdom Medical Research Council programme (grant MC UU 00002/2) and theme (grant MC_UU_00040/02 – Precision Medicine) funding. LB is supported by grants from Kennedy Trust for Rheumatology Research (grant number 171806) and UK Medical Research Council (grant MC UU 00002/2). This work was supported by the NIHR Oxford Biomedical Research Centre (BRC) and the NIHR Nottingham BRC. The views expressed are those of the authors and not necessarily those of the NHS, the NIHR or the Department of Health.

## Author contributions

Conception and Design: TLV, FEW, LJD, PAH, RAM, JG, SL, SB, LSL, AS, CTA, SK, NKA, DF, BDMT, MK, TJW, DAW, AMV. Analysis and interpretation of data: TAP, YD, LJD, FEW, TLV, PAH, RAM, JM, SB, BDMT, LB. Drafting Article: TAP, TLV, YD, LJD, FEW, BDMT. Critical revision of article: all authors. Final Approval: all authors.

## Competing interests

TAP, YD, PAH, SL, AS, NKA, DF, MK, AMV, BDMT, LB and SK declare no conflicts of interest. FEW has received consultancy fees from Pfizer and Novartis. LSL has received consultancy fees from Arthro Therapeutics AB, and was an advisory board member of AstraZeneca. LJD has received consultancy fees from Nightingale Health PLC. TLV has no conflicts to declare with the exception of grant income for STEpUP OA from industry partners (see above). RAM is a shareholder of AstraZeneca. SB and JM are employees and shareholders of Novartis. JG is an employee and shareholder of Standard BioTools (formally SomaLogic). CTA has received consultancy fees from Novartis, and has received honoraria for educational purposes also from Novartis. TJW is a shareholder of Chondropeptix BV. DAW has received consultancy fees from GlaxoSmithKline plc, AKL Research & Development Limited, Pfizer Ltd, Eli Lilly and Company, Contura International, and AbbVie Inc, has received honoraria for educational purposes from Pfizer Ltd and AbbVie Inc, is a board member (Director) of UKRI and Versus Arthritis Advanced Pain Discovery Platform. The authors declare no other competing interests.

## Additional information

## the STEpUP OA Consortium

**University of Oxford** Thomas A. Perry[1], Yun Deng[1], Philippa A. Hulley[2], Rose M. Maciewicz[1], Stefan Kluzek[2,7], Nigel K. Arden[2,8], Luke Jostins-Dean ©[1,22], Tonia L. Vincent[1,22], Vicky Batchelor[1], Jennifer Mackay-Alderson[1], Gretchen Brewer[1], Brian Marsden[2], Andrew J. Price[2], Megan Goff[1], Vinod Kumar[1], James Tey[2] & Tamas Szommer[1]

**Novartis** Joanna Mitchelmore[3], Sophie Brachat[3], Juerg Gasser[3] & Lori Jennings[3]

**Lund University** Staffan Larsson[4], André Struglics[4] & L. Stefan Lohmander[4]

**Standard BioTools (formally SomaLogic)** Joe Gogain[5], Darryl Perry[5], Anna Mitchel[5] & Ela Zepko[5]

**University of Western Ontario** C. Thomas Appleton[6], Trevor B. Birmingham[6] & J. Daniel Klapak[6]

**Boston University** David Felson[9]

**University of Cambridge** Laura Bondi[10] & Brian D. M. Tom[10]

**University of Toronto** Mohit Kapoor[11], Rajiv Gandhi[11], Anthony Perruccio[11], Y. Raja Rampersaud[11] & Kim Perry[11]

**University College Maastricht** Tim J. Welting[12], Pieter Emans[12], Tim Boymans[12], Liesbeth Jutten[12], Marjolein Caron[12] & Guus van den Akker[12]

**University of Nottingham** David A. Walsh[13,14], Ana M. Valdes[13], Michael Doherty[13] & Vasileios Georgopoulos[13]

**Imperial College London** Fiona E. Watt ⓡ [1,15,22] & Artemis Papadaki[15]

**Fortius Clinic** Andrew Williams[16]

**University of Manchester** Tim Hardingham[17]

**Biosplice** Sarah Kennedy[18] & Jeymi Tambiah[18]

**Fidia** Devis Galesso[19] & Nicola Giordan[19]

**UCB** Waqar Ali[20]

[16]Fortius Clinic, London, UK. [17]Division of Cell-Matrix Biology and Regenerative Medicine, Wellcome Trust Centre for Cell-Matrix Research, Faculty of Biology, Medicine and Health, School of Biological Sciences, University of Manchester, Manchester, UK. [18]Biosplice Therapeutics, Inc., 9360 Towne Centre Dr, San Diego, CA, USA. [19]Fidia Farmaceutici S.p.A, 35031 Abano Terme, Italy. [20]UCB Pharma UK, Slough, UK.

