## [Transparent Peer Review file · Nature Communications]

Large-scale molecular endotype discovery in synovial fluid reveals osteoarthritis as a single biological continuum

Corresponding Author: Professor Tonia Vincent

Version 0:

Reviewer comments:

Reviewer #1

(Remarks to the Author)

This revised version of the manuscript together with the explanatory text in the rebuttal substantially addresses my queries arising from the original manuscript. My single comment that remains insufficiently addressed is as follows:

Reviewer 3 comment 3, text edit response page 14 lines 411-413: This statement of limitation does not actually acknowledge the limitation, but makes it sound more like a strength. This is insufficient. Please describe the limitation of SF sampling opportunity explicitly as a limitation rather than camouflage it with percentage success data from a single cohort representing a small subset of the total. If you cannot give the sampling success for all of the cohorts, you should not mention it for a single cohort, as this gives a skewed perspective of the general feasibility of the method.

(Remarks on code availability)

Reviewer #2

(Remarks to the Author)

The proteomic assay of the synovial fluid from 1361 patients was used to determine if there were observable molecular features that would associate with endotypes as defined by variation in clinical features. The data show however that OA is not able to be associated with unique endotypes rather that it is characterized as biological continuum that was associated with an EMT transition ontology that was relatable to disease severity as defined by radiographic disease severity.

Within in this data set besides WOMAC did the authors have information to also stratify the data into those that went onto to surgical treatment to both WOMAC and severity to the DE in the SF.

More interesting none of these features related to pain suggesting that some other type of biological or psychological processes drove pain. A more extensive discussion relative to the relationship between pain and treatment should be made. It is well known that severity by physical signs does not relate to pain and patients undergo total knee replacement because pain interferes with function. The lack of relationship between WOMAC and the SF proteome should be more specifically be elaborated in the discussion

(Remarks on code availability)

Reviewer #3

(Remarks to the Author)

The authors have address the critiques from my prior review.

(Remarks on code availability)

Version 1:

Reviewer comments:

Reviewer #1

(Remarks to the Author)

The authors have thoroughly addressed all of the reviewer comments and have been very responsive to both addressing the various critiques and suggestions for improvements to the manuscript.

(Remarks on code availability)

Dear Editor,

We very much appreciate the efforts to process our manuscript swiftly and have pleasure in returning an edited version which addresses the final reviewer comments. See below.

Yours sincerely,

Reviewer #1 (Remarks to the Author):

This revised version of the manuscript together with the explanatory text in the rebuttal substantially addresses my queries arising from then original manuscript. My single comment that remains insufficiently addressed is as follows:

Reviewer 3 comment 3, text edit response page 14 lines 411-413: This statement of limitation does not actually acknowledge the limitation, but makes it sound more like a strength. This is insufficient. Please describe the limitation of SF sampling opportunity explicitly as a limitation rather than camouflage it with percentage success data from a single cohort representing a small subset of the total. If you cannot give the sampling success for all of the cohorts, you should not mention it for a single cohort, as this gives a skewed perspective of the general feasibility of the method.

Author Response: We do have this figure available on 4 of the larger cohorts in STEpUP OA, which account for 51% of the total OA SF sample number. We acknowledge that this remains a limitation and generalisability to the broader OA population is unknown. We hope that our wording now reflects this.

Author Action: The following edits have been made to the discussion (pg.14, lines 419-424):

*“Our samples were generated from a diverse set of, largely, pre-existing cohorts. The **percentage** who had successful SF aspiration **was documented in only 4/17 of the cohorts (albeit accounting for 51% of the total participant number)**. In these instances, successful aspiration of SF was greater than 65%, **but it remains unclear how representative this is of the whole cohort and whether this may have biased the biology revealed in our analysis and the generalisability of OA**. Our analysis was powered to find several endotypes within the whole OA population and to detect two distinct endotypes if considering non-advanced radiographic disease only”.*

Reviewer #2 (Remarks to the Author):

The proteomic assay of the synovial fluid from 1361 patients was used to determine if there were observable molecular features that would associate with endotypes as defined by variation in clinical features. The data show however that OA is not able to be associated with unique endotypes rather that it is characterized as biological continuum that was associated with an EMT transition ontology that was relatable to disease severity as defined by radiographic disease severity.

Within in this data set besides WOMAC did the authors have information to also stratify the data into those that went onto to surgical treatment to both WOMAC and severity to the DE in the SF.

Response: The analysis presented in this primary analysis is only cross-sectional. We do have some patients/cohorts that have prospective data beyond the synovial fluid sampling date. Around 311 patients have recorded follow up clinical data (2-5 years). This is largely a change in clinical outcome

measure (WOMAC) and generally does not include surgical outcome (very small numbers where this is the case). Subsequent analyses will focus on this in depth with their relationship with SF signature. It is beyond the remit of the current study.

Author Action: We have stressed the cross-sectional nature of our pain findings and indicate that a smaller scale prospective study will be performed in due course. The text has been edited as below.

More interesting none of these features related to pain suggesting that some other type of biological or psychological processes drove pain. A more extensive discussion relative to the relationship between pain and treatment should be made. It is well known that severity by physical signs does not relate to pain and patients undergo total knee replacement because pain interferes with function. The lack of relationship between WOMAC and the SF proteome should be more specifically be elaborated in the discussion.

Author Response: The reviewer is right in drawing attention to the fact that there were a no robust (replicable) molecular markers associated with pain, which is likely to be related to several factors. Perhaps the most likely is the unreliability of using self-reported pain measures in a cross-sectional study of this sort. Other factors include medications that the patient may have been on and other co-morbidities/factors that contribute to the experience of pain. We have expanded this section of the discussion to reflect this further.

Author Action: The following paragraph has been edited to cover the points raised above (pg.13, lines 398-408).

“Replication in STEpUP OA was robust for associations with structural disease but less so for pain. This lack of association is unlikely to be because the joint does not contain molecules that are directly involved in triggering pain responses, as most individuals (>80%) gain symptomatic benefit after joint replacement. It seems more likely that this is due to high variability in patient reported symptomatic outcomes which are known to be influenced by external factors beyond molecular drivers made by the joint e.g. psychological factors^[59], biological sex. This makes cross-sectional analyses of this sort challenging. Levels of pain will also be influenced by analgesic treatments that the patient was taking, although this was not captured comprehensively across the whole cohort and was not therefore included as a potential confounding factor. Only a small proportion of the individuals in STEpUP OA had associated prospective clinical outcome data (mainly pain scores). These were not included in the current reported study but will be examined separately in future work”.

Reviewer #3 (Remarks to the Author):

The authors have address the critiques from my prior review.

Author Response/Action: Nil to add.